# *Canavalia gladiata* Pod Extract Mitigates Ovalbumin-Induced Asthma Onset in Male BALB/c Mice via Suppression of MAPK

**DOI:** 10.3390/molecules27196317

**Published:** 2022-09-25

**Authors:** Wen Yan Huang, Sang Hoon Lee, Seong Ju Oh, Hyeock Yoon, Jeong Hoon Pan, Inhye Jeong, Mi Jeong Kim, Bok Kyung Han, Jae Kyeom Kim, Eui-Cheol Shin, Young Jun Kim

**Affiliations:** 1Department of Food and Biotechnology, Korea University, Sejong 30019, Korea; 2BK21 FOUR Research Education Team for Omics-Based Bio-Health in Food Industry, Korea University, Sejong 30019, Korea; 3Department of Food and Regulatory Science, Korea University, Sejong 30019, Korea; 4Department of Food and Nutrition, Chosun University, Gwangju 61452, Korea; 5RWJMS Institute for Neurological Therapeutics, Rutgers-Robert Wood Johnson Medical School, Piscataway, NJ 08854, USA; 6Department of Behavioral Health and Nutrition, College of Health Sciences, University of Delaware, Newark, DE 19716, USA; 7Department of Food Science, Gyeongsang National University, Jinju 52828, Korea

**Keywords:** *Canavalia gladiata* pod, antiallergy, anti-inflammation, lung damage, ethanolic extract

## Abstract

Asthma is one of the most common inflammatory diseases of the lung worldwide. There has been considerable progress in recent studies to treat and prevent allergic asthma, however, various side effects are still observed in clinical practice. Six-week-old male BALB/c mice were orally administered with either sword bean pod extracts (SBP; 100 or 300 mg/kg) or dexamethasone (DEX; 5 mg/kg) once daily over 3 weeks, followed by ovalbumin sensitization (OVA/Alum.; intraperitoneal administration, 50 μg/2 mg/per mouse). Scoring of lung inflammation was performed to observe pathological changes in response to SBP treatment compared to OVA/Alum.-induced lung injury. Additionally, inflammatory cytokines were quantified in serum, bronchoalveolar lavage fluid (BALF), and lung tissue using ELISA and Western blot analyses. SBP treatment significantly reduced the infiltration of inflammatory cells, and release of histamine, immunoglobulin E, and leukotriene in serum and BALF. Moreover, the therapeutic effect of SBP was also assessed to analyze the inflammatory changes in the lung tissues. SBP markedly suppressed the activation of the MAPK signaling pathway and the expression of key inflammatory proteins (e.g., TNF-α) and Th2 type cytokines (IL-5 and IL-13). SBP was effective in ameliorating the allergic inflammation against OVA/Alum.-induced asthma by suppressing pulmonary inflammation.

## 1. Introduction

Asthma is one of the most common chronic respiratory diseases affecting 300–400 million people worldwide [1]. The global incidence of asthma is increasing every year due to air pollution [2]. The imbalance in the ratio of Th1 to Th2 cells is the major cause of asthma [3], with regard to overproduction of Th2 cytokines (e.g., interleukin (IL)-5 and IL-13) and histamine, allergen-specific immunoglobulin (i.e., IgE), and leukotriene [4,5]. In addition, several recent reports have revealed that mitogen-activated protein kinases (MAPKs) play an important role in pathogenesis of asthma [6,7,8]. In a clinical study, airway biopsy samples from allergic athematic patients demonstrated increased MAPK activation, which was correlated with the clinical severity and intensity of asthma [9]. Thus, inhibition of MAPKs could be an emerging target to decrease inflammation and remodeling in asthma.

Systemic corticosteroids are widely used both intermittently and long-term as a treatment for this but can cause multiple side effects [10,11]. Natural products (e.g., herbs) are considered alternatives of the medications with side effects [12,13]. *Canavalia gladiata*, also known as sword bean (SB), is a leguminous annual plant originating from Asia and Africa [14]. SB has been used as a traditional medicine for inflammatory diseases in Asian countries including Korea, China, and Japan [15]. In addition to SB, pods of the SB are also promising materials due to their multifactorial activities (e.g., antioxidative, antibacterial, and anti-inflammatory therapeutic activities) [11,16]. For instance, it was reported that the pod of SB showed anti-inflammatory effects in LPS-induced RAW264.7 cells [16]. Additionally, it was demonstrated that the anti-inflammatory effect of the pod of SB was linked to regulating Th1/Th2 cell differentiation in an allergy model [17].

Despite the known health benefits of SB pods (SBPs), there is limited scientific evidence available for the preventive effects of SBP on asthma. Therefore, this study aims to examine the preventive effects of ethanolic extract of SBP on chemically induced asthma and related mechanisms. First, the chemical profile (HPLC analysis) and antioxidant capacity (radical scavenging ability) of SBP were characterized in in vitro levels. In addition, ovalbumin (OVA) adsorbed in aluminum hydroxide (Alum.) (OVA/Alum.) was utilized to induce asthma conditions in mice, and dose-dependent effects of SBP were assessed against OVA/Alum.-sensitized asthma mice by measuring the histological change and related biochemical analysis (ELISA and Western blot analysis). 

## 2. Results and Discussion

### 2.1. Free Phenolic Acid, Antioxidant Capacity, and Phytochemicals in SBP

It is well known that antioxidant capacity and polyphenol content are positively associated, which is beneficial to human health. In that regard, in vitro antioxidant capacity, total polyphenol content (TPC), and total flavonoid content (TFC) of SBP were examined (Table 1). SBP showed 31.61 ± 0.19 mg vitamin C equivalent (VCE)/g and 20.61 ± 1.06 mg VCE/g in ABTS and DPPH assays, respectively. TPC and TFC of SBP were 16.52 ± 0.21 mg gallic acid equivalent (GAE)/g and 6.82 ± 0.13 mg of rutin equivalent (RE)/g, respectively. Subsequently, representative polyphenols in SBP were identified using HPLC analysis. Specifically, a total of nine phenolic compounds (i.e., gallic acid, p-hydroxybenzoic acid, methyl gallate, vanillic acid, caffeic acid, ferulic acid, sinapic acid, benzoic acid, and salicylic acid) were selected based on a previous report (Figure 1) [16]. Of the nine polyphenols, five polyphenols were detected in our extraction condition. Gallic acid (114.03 ± 0.83 µg/g) was the most abundant, followed by ferulic acid (24.63 ± 0.11 µg/g), caffeic acid (21.29 ± 0.11 µg/g), p-hydroxybenzoic acid (15.90 ± 0.14 µg/g), and vanillic acid (15.16 ± 0.04 µg/g) (Table 2). Several studies reported that gallic acid and vanillic acid mitigated the OVA/Alum.-induced asthma in animal models through prevention of airway inflammation [18,19]. Lee et al. reported an antiallergic effect of ferulic acid in an asthmatic mouse model, which is via restoring Th1/Th2 imbalance by modulating dendritic cell function [20]. In particular, gallic acid, with the highest content measured in SBP, is known to be effective in treating asthma through regulating the factors and cytokines (IL-5, IL-13, etc.) related to the MAPK pathway [6,21]. Moreover, caffeic acid phenethyl ester alleviated asthma by regulating the airway microenvironment via the ROS-responsive MAPK/Akt pathway [22].

### 2.2. Effects of SBP on OVA/Alum.-Induced Pulmonary Damage

The protective effect of SBP against OVA/Alum.-induced lung injury was examined via histopathological analysis of lung tissue sections. H&E-stained tissue slides were utilized for injury scoring. As shown in Figure 2, OVA/Alum. challenge induced more inflammatory cell infiltration in cells located on the alveolar wall, which was lower by 10% in SBP mice although there was no dose-dependent effect. As expected, DEX significantly attenuated the inflammation score compared to NC mice. Of note, multiple studies demonstrated that lung injury markers are increased in response to OVA/Alum. challenge [23,24]. Similarly, those lung injury markers were elevated in NC mouse lung tissues in the present study. SBP alleviated the injury markers in lung tissues (e.g., morphological changes, infiltration of inflammatory cells in lung tissue, and further release of histamine).

### 2.3. Effects of SBP on OVA/Alum.-Induced Inflammation

Representative proinflammatory mediators (i.e., histamine, IgE, and leukotriene) in serum and BALF were analyzed. Serum histamine and IgE were significantly increased in NC mice, which were reversed in PC and SBP mice, but there was no dose-dependent effect of SBP (Figure 3A,B). Leukotriene level was measured in serum and BALF. Leukotriene in BALF was dramatically increased by OVA/Alum. challenge, which was decreased in PC and SBP 300 mice (Figure 3D). There was a trend of increase in the serum leukotriene in NC mice (Figure 3C). Serum leukotriene level in PC and SBP mice was as low as Normal mice, but there was no statistical difference with NC mice (Figure 3C). It is well established that OVA/Alum. sensitization increases in the levels of inflammatory mediators, histamine, and IgE in serum and infiltration of inflammatory cells in lung tissues [25]. Additionally, additional OVA challenge enhances allergic reaction. Meanwhile, IgE molecules bind to mast cell receptors, thereby increasing release of the inflammatory mediators via mast cell degranulations [26,27]. The released IgE is responsible for increased production of leukotrienes, which in turn causes increased allergic responses via attracting eosinophils [28]. In our study, SBP suppressed the release of serum IgE and BALF leukotriene compared to NC mice, indicating SBP is beneficial to alleviate OVA/Alum.-induced allergic responses.

### 2.4. Effects of SBP Treatment on OVA/Alum.-Induced Signaling Pathways

Key inflammatory proteins were measured via Western blot analysis. MAPK signaling modules were divided into three groups: extracellular signal-regulated kinases (ERKs), P38 MAPKs, and c-Jun N-terminal kinases (JNKs), which play important roles in the expression and activation of inflammatory mediators in the airways [6,29]. Phosphorylation of ERK, JNK, and P38 was significantly increased in NC mice compared to the Normal mice, whereas phosphorylated ERK, JNK, and P38 in SBP mice were as low as PC mice (Figure 4). Activation of the intracellular MAPK signaling pathway underlies the inflammatory processes of asthma. For instance, ERK and JNK activation increases inflammatory cytokines including IL-5 and IL-13 [30]. In addition, p38 activation is accompanied by cytokine upregulation in macrophages of asthmatic patients [30]. Collectively, inhibition of the MAPK signaling pathway could be a therapeutic target for allergic asthma [30,31]. 

In addition to the MAPK signaling pathway, key inflammatory proteins (i.e., TNF-α, iNOS, and IL-6) and Th2 type cytokines (IL-5 and IL-13) were also examined. All the five proteins were significantly upregulated by OVA/Alum. challenge (Figure 5). Overall, PC and SBP reduced the five proteins as low as Normal mice although SBP 300 and SBP 100 did not change iNOS and IL-13, respectively. Allergen exposure in airways causes increased Th2 type cytokines (i.e., IL-4, IL-5, and IL-13), leading to activation of the Th2 immune response [32]. For example, IL-5 plays a crucial role in eosinophil maturation and recruitment, and IL-13 leads to B-cell activation, IgE hypersecretion, goblet cell formation, and mucosal overproduction [33]. IL-13 also induces the expression of iNOS in human bronchial epithelial cells and activates macrophages [33,34], indicating that Th2 cytokines are closely related to signaling pathways of asthma onset. Taken together, protective effects of SBP against the OVA/Alum.-induced inflammatory mechanism in asthma onset might be via suppression of Th2 cytokines.

## 3. Materials and Methods

### 3.1. Materials and Reagents

Dried SB pods were purchased at a local grocery store (Hwasun, Jeollanam-do, South Korea). Powdered SB pods were extracted twice with 30% ethanol (1:10, *v/v*) at 80 °C for 8 h (second extraction for 4 h). The obtained extract was filtered, concentrated, spray dried, and stored at −20 °C before use [17].

### 3.2. Analysis of Free Phenolic Acid

The free phenolic acid content of SBP was analyzed by a slightly modified method [16]. One gram of dried SBP was extracted three times with 20 mL of 70% methanol–70% acetone (1:1, *v/v*). After centrifugation, the supernatant was evaporated at 45 °C to 20 mL, and then the aqueous suspension was adjusted to pH 2.0 with 5 N HCl and extracted with hexane (1:1, *v/v*). The water phase was extracted three times with diethyl ether/ethyl acetate (DE/EA 1:1, *v/v*), and dehydrated, filtered, and evaporated to dryness under nitrogen gas. Dried samples were dissolved in 80% methanol/water and filtered (0.2 µm disposable filter, Whatman, MA, USA). For the HPLC analysis, the analytical HPLC system employed comprised the Shiseido Nanospace SI-2 series (Shiseido, Tokyo, Japan) and a photodiode array detector (Accela; Thermo Fisher Scientific, Waltham, MA, USA) was used. Phenolic acid separation was carried out by an analytical column (Shiseido, Capcellpak C18 UG 120 5 µm, 4.6 × 250 mm). Gradient elution was employed with a mobile phase consisting of 0.1% formic acid in distilled water (solution A) and 0.1% formic acid in acetonitrile (solution B) as follows: isocratic elution 5% B at 0–5 min; linear gradient from 5% to 30% B at 5–55 min; 30–95% B at 55–60 min. The composition was next held at 95% B for 5 min, and then it returned to its initial conditions and was maintained for 5 min to equilibrate the column before the next injection. The detection wavelength was 270 nm. The flow rate was 1.0 mL/min, and the injection volume was 10 µL.

### 3.3. Antioxidant Capacity and Component Assay

#### 3.3.1. ABTS Radical Scavenging Assay

A solution of cation-radical ABTS was prepared according to Shalaby and Shanab [35] with some modifications. Briefly, 190 μL of ABTS (1 mM AAPH and 2.5 mM ABTS in PBS; Sigma-Aldrich, St. Louis, MO, USA) solution was mixed with 10 μL of SBP (1 mg/mL; distilled water (DW)) solution and allowed to react for 10 min at room temperature in the dark. Absorbance at 734 nm was measured using a spectrophotometer (SPECTROstar Nano; BMG LABTECH Corp., Ortenberg, Germany). The radical scavenging activity of the SBP was expressed as the ascorbic acid equivalent antioxidant capacity (VCE).

#### 3.3.2. DPPH Radical Scavenging Assay

The DPPH radical scavenging activity was measured according to the modified method of Shalaby and Shanab [35]. Briefly, 190 μL of DPPH (50 μM in 95% ethanol; Sigma-Aldrich, St. Louis, MO, USA) solution was added to 10 μL of SBP (1 mg/mL; 95% EtOH) solution and left in the dark at room temperature. After 30 min, the absorbance of samples was measured at 517 nm using a spectrophotometer (LABTECH Corp., Ortenberg, Germany) and the results are presented as the mg VCE mg/g sample.

#### 3.3.3. Total Polyphenol Content

The total polyphenol content (TPC) of SBP was determined by testing on a laboratory scale with some modifications in volume and concentration referring to the method of Singh et al. [36]. Briefly, 50 μL of the SBP solution (1 mg/mL; DW) was reacted with 100 μL of Na_2_CO_3_ (2% in DW; Sigma-Aldrich, St. Louis, MO, USA) solution and 50 μL of Folin–Ciocalteu solution (0.2 N in DW; Sigma-Aldrich, St. Louis, MO, USA), mixed, and left in the dark. After 10 min, the absorbance was measured at 720 nm using a spectrophotometer (LABTECH Corp., Ortenberg, Germany), and the results are presented as the gallic acid equivalence value (GAE).

#### 3.3.4. Total Flavonoid Content

The total flavonoid content (TFC) of SBP was measured by adjusting the volume and concentration in Singh et al. [36]. In brief, the 500 μL SBP solution (1 mg/mL) was mixed with 200 μL NaNO_2_ (10% in DW; Sigma-Aldrich, St. Louis, MO, USA), and the mixture was left in the dark at room temperature. After 6 min, 300 μL of AlCl_3_ (10% in DW; Sigma-Aldrich, St. Louis, MO, USA) was added and incubated for 6 min again, then 2 mL of NaOH (1 M in DW; Sigma-Aldrich, St. Louis, MO, USA) was added, blended, and left for 15 min at room temperature. The absorbance was measured at 510 nm using a spectrophotometer (LABTECH Corp., Ortenberg, Germany), and the results are presented as the rutin equivalence value (RE).

### 3.4. Animals and Treatment

Thirty-five 6-week-old male BALB/c mice (18–20 g) were purchased from Raon Bio Inc. (Yong-in, Korea) and housed in a specific pathogen-free environment on a 12/12 h light/dark cycle at 22–25 °C with 50–60% relative humidity. After adaptation for one week, the mice were randomly divided into five groups (each with 7 mice): normal control (Normal; saline), negative control (NC; OVA/Alum. (Sigma-Aldrich, St. Louis, MO, USA)-sensitized + saline), positive control (PC; OVA/Alum.-sensitized + dexamethasone (DEX; Sigma-Aldrich, St. Louis, MO, USA) at 5 mg/kg); SBP100 (OVA/Alum.-sensitized + SBP at 100 mg/kg); and SBP 300 (OVA/Alum.-sensitized + SBP at 300 mg/kg). Saline, DEX, and SBP were orally administered once daily for three weeks. For initial sensitization, all mice, except those in the Normal group, were intraperitoneally injected with a mixture of 50 μg of OVA and 2 mg of Alum. in 200 μL of saline on day 1 and day 14. From days 16 to 20, after oral treatment for 1 h, the mice were subjected to 1% (*w/v*) OVA nebulization for 30 min daily. The mice in the Normal groups were nebulized with an equal volume of saline. After the last dosing on day 21, all mice were euthanized by an intraperitoneal injection of tribromoethanol (200 mg/kg; Sigma-Aldrich, St. Louis, MO, USA), and their serum, bronchoalveolar lavage fluid (BALF), and lung tissues were immediately collected and stored at −80 °C. All animal care and experimental procedures were approved by the ethics committee of Korea University (approval number: KUIACUC-2022-0052).

### 3.5. Histological Analysis

The left lobe of the lung tissue was collected and fixed in 4% paraformaldehyde, then embedded in paraffin and cut into 3 μm sections. The areas of inflammation and injury spots were analyzed with hematoxylin and eosin (H&E; Sigma-Aldrich, St. Louis, MO, USA) staining. The stained images were visualized under an inverted microscope (Olympus BH 2, Tokyo, Japan; magnification, ×100). The lung inflammation score was measured on a subjective scale of 1 to 3 as follows: 1, no inflammation was observed; 2, there was occasional cuffing with inflammatory cells; and 3, most bronchi or vessels were surrounded by inflammatory cells [37,38].

### 3.6. ELISA Assay

The levels of histamine (ab213975; abcam, Cambridge, UK), IgE (ab157718; abcam, Cambridge, UK), and leukotrienes (ab133042; abcam, Cambridge, UK) in serum and BALF were measured using specific ELISA kits according to the manufacturer’s instructions.

### 3.7. Immunoblot Analysis

The collected lung tissue was lysed in a radioimmunoprecipitation assay buffer (Sigma-aldrich, St. Louis, MO, USA) with a phosphatase inhibitor cocktail from Roche (Mannheim, Germany) for 20 min at 4 °C. After the centrifugation at 12,000× *g* at 4 °C, the supernatants were obtained. Subsequently, equal amounts of each protein sample (20 μg) were separated by sodium dodecyl salt-polyacrylamide gel electrophoresis and transferred onto a membrane of polyvinylidene difluoride (Millipore, Boston, MA, USA). It was blocked with 5% skim milk solution at room temperature for one hour, and incubated overnight with the primary antibodies (i.e., β-actin, p-ERK, ERK, p-JNK, JNK, p-P38, P38, iNOS, TNF-α, IL-5, IL-6, and IL-13) (Cell Signaling, Danvers, MA, USA; Santa Cruz Biotechnology, Dallas, TX, USA; Abcam, Cambridge, UK)) at 1/1000 or 1/2000 dilutions with agitation at 4 °C. The blots were incubated with a secondary antibody for 1 h at room temperature, visualized using enhanced chemiluminescence reagents (Bio-Rad Laboratories, Hercules, CA, US), and examined with an Image Quant LAS-4000 chemiluminometer (GE Healthcare, Chicago, IL, USA). 

### 3.8. Statistical Analysis

The experimental data were processed using IBM SPSS software (version 25.0; IBM Corp, Armonk, NY, USA). Unless otherwise specified, the data are expressed as mean or from three independent experiments. The differences between the groups were analyzed with a one-way analysis of variance followed by Dunnett’s multirange test, where * *p* < 0.05, ** *p* < 0.01, and *** *p* < 0.001 were considered to indicate a significant difference.

## 4. Conclusions

We herein assessed the protective effects of SBP on OVA/Alum.-induced asthma onset using a mouse model. In most cases, SBP mitigated OVA/Alum.-induced lung injury via suppressing MAPK activation and inflammatory cytokines, hence anti-inflammatory. The beneficial effects of SBP could be antioxidative properties of its abundant polyphenols, which were identified and quantified in the present study. However, there are limitations to be mentioned for this study. Even though we used two different doses of SBP, there was no difference between 100 mg/kg and 300 mg/kg of SBP. It is likely that SBP reached a plateau showing its maximum efficacy at 100 mg/kg, which can be considered a strength of SBP since 100 mg/kg of the extract is an easily achievable dose. Thus, current findings warrant a further study on the effect of lower doses of SBP for more economical applications. Additionally, combined use of SBP with asthma medications can be considered after establishing additional study on adverse effects since SBP showed strong effects on asthma onset. The combined use of SBP with asthma drugs could be beneficial in terms of cost saving and lowering side effects by lowering the dose of drugs.

## Figures and Tables

**Figure 1 molecules-27-06317-f001:**
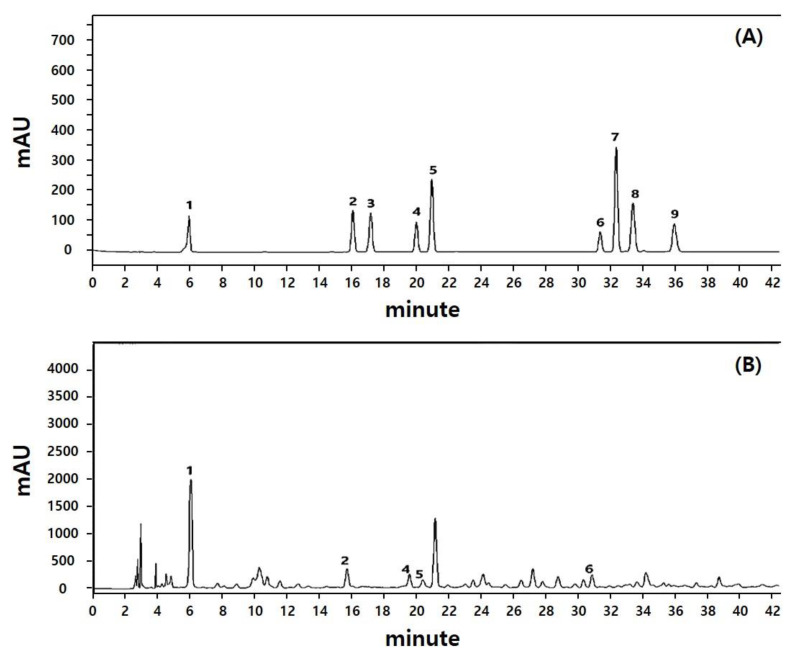
HPLC chromatograms of the phenolic acid standard mixture (**A**) and profiles of SBP extract (**B**). 1. gallic acid, 2. p-hydroxybenzoic acid, 3. methyl gallate, 4. vanillic acid, 5. caffeic acid, 6 ferulic acid, 7. sinapic acid, 8. benzoic acid, 9. salicylic acid.

**Figure 2 molecules-27-06317-f002:**
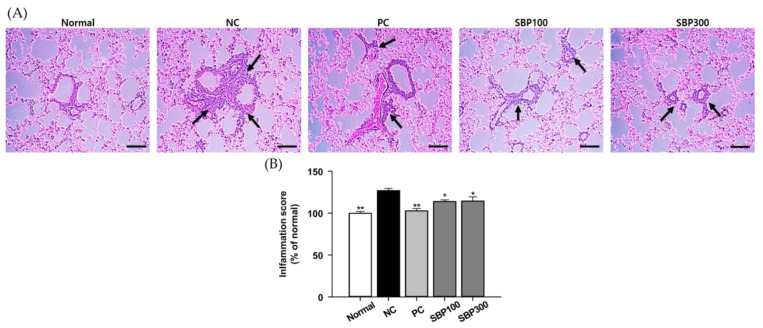
Histological analysis of representative hematoxylin and eosin-stained lung sections, and lung pathology assessment of OVA-induced lung damage. (**A**) Hematoxylin and eosin-stained sections, the arrow indicates the area of cell infiltration. (**B**) inflammation score quantification. Normal, normal control; NC, negative control; PC, positive control; SBP100, SBP at 100 mg/kg; SBP300, SBP at 300 mg/kg. All values are expressed as mean ± SEM (*n* = 7). *p*-value less than 0.05 was considered statistically significant. * *p* < 0.05, and ** *p* < 0.01 indicate statistical significance compared to the NC group.

**Figure 3 molecules-27-06317-f003:**
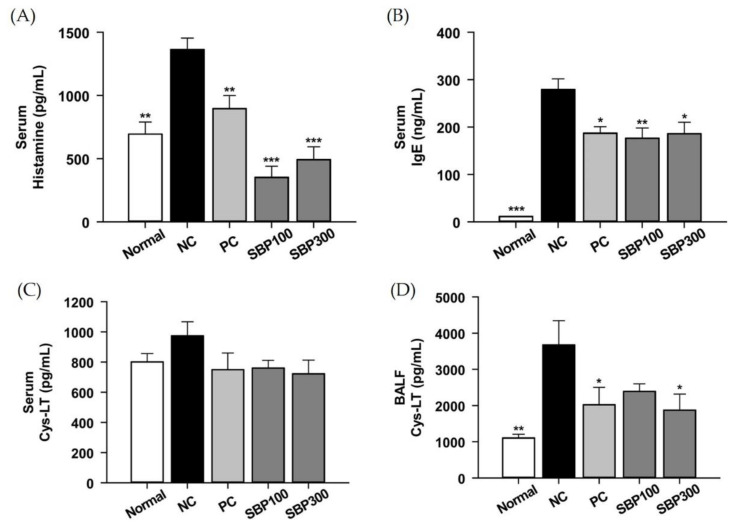
The expression of proinflammatory cytokines in the serum (**A**–**C**) and BALF (**D**) from OVA-treated mice. (**A**) Histamine, (**B**) IgE protein expression. Normal, normal control; NC, negative control; PC, positive control; SBP100, SBP at 100 mg/kg; SBP300, SBP at 300 mg/kg. All values are expressed as mean ± SEM (*n* = 7). *p*-value less than 0.05 was considered statistically significant. * *p* < 0.05, ** *p* < 0.01, and *** *p* < 0.001 indicate statistical significance compared to the NC group.

**Figure 4 molecules-27-06317-f004:**
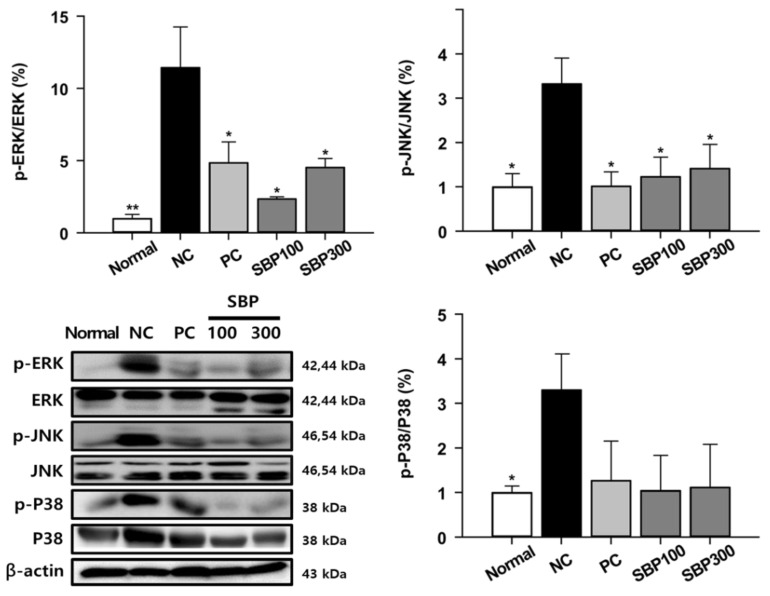
The expression of inflammatory proteins in the lung tissue from OVA-treated mice. Normal, normal control; NC, negative control; PC, positive control; SBP100, SBP at 100 mg/kg; SBP300, SBP at 300 mg/kg. All values are expressed as mean ± SEM (*n* = 7). *p*-value less than 0.05 was considered statistically significant. * *p* < 0.05 and ** *p* < 0.01 indicate statistical significance compared to the NC group.

**Figure 5 molecules-27-06317-f005:**
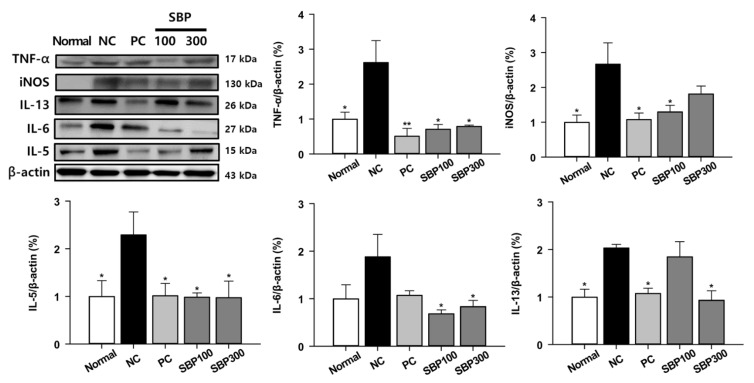
The expression of inflammatory cytokines in the lung tissue from OVA-treated mice. Normal, normal control; NC, negative control; PC, positive control; SBP100, SBP at 100 mg/kg; SBP300, SBP at 300 mg/kg. All values are expressed as mean ± SEM (*n* = 7). *p*-value less than 0.05 was considered statistically significant. * *p* < 0.05 and ** *p* < 0.01 indicate statistical significance compared to the NC group.

**Table 1 molecules-27-06317-t001:** Radical scavenging capacity and phytochemicals in sword bean pod.

ABTS Assay	DPPH Assay	Total Polyphenol Content	Total Flavonoid Content
mg VCE/g sample		mg GAE/g sample	mg RE/g sample
31.61 ± 0.19	20.61 ± 1.06	16.52 ± 0.21	6.82 ± 0.13

Data are expressed as the mean ± SD (*n* = 3). VCE, vitamin C equivalent; GAE, gallic acid equivalent; RE, rutin equivalent.

**Table 2 molecules-27-06317-t002:** The content of free phenolic compound in SBP extract.

Gallic Acid	p-Hydroxybenzoic Acid	Vanillic Acid	Caffeic Acid	Ferulic Acid
µg/g sample (retention time (min))				
114.03 ± 0.83 (5.9)	15.90 ± 0.14 (15.7)	15.16 ± 0.04 (19.5)	21.29 ± 0.11 (20.5)	24.63 ± 0.11 (30.9)

Data are expressed as the mean ± SD (*n* = 3).

## Data Availability

Not applicable.

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
