# Peer review of "Canavalia gladiata Pod Extract Mitigates Ovalbumin-Induced Asthma Onset in Male BALB/c Mice via Suppression of MAPK"

_molecules, 2022, doi:10.3390/molecules27196317_

Round 1

Reviewer 2 Report

Comments:

Line 55 – The author put the abbreviation of the Latin name in brackets, which is redundant after the full Latin name;

At the end of the introduction, the authors should better explain undertaken experiments to solve the problem.

Line 72 – antioxidant, instead of ani-oxidant;

Line 75 – What does VCE/g mean? The author should explain abbreviations before writing them;

Line 86 – What is antioxidant disease? If it is oxidative stress, you should write it exactly;

Line 90 – If the author examined the profile and content of phenolic acids, then he should better discuss their relationship with asthma or asthma markers;

Line 142 and next – The author cannot compare results between Normal, NC, PC, SBP100, and SBP300 samples because they calculated the Dunnet's test, which allows comparison to the control only;

Therefore, they should calculate another test (Tuckey, Scheffe, or nonparametric) which allows multiple comparisons.

Line 181 and next – What was the mixture used for phenolics extraction from the water phase? There is (line 186) diethyl ether - ethyl (DE/EA 1:1, v/v). Please describe it precisely;

What detector was used?

What data were used for phenolics determination?

Line 205 – Please specify spectrophotometer;

Line 224 – Please explain what the slight modification was; 

Line 239 – In the materials and methods section, the author describes two samples, SBP low and SBP high, whereas, in the results section, the author shows SBP100 and SBP300. Please use the same abbreviations in different parts of the manuscript.

Line 260 – Please clearly specify what kits were used for ELISA assays.

Round 2

Reviewer 1 Report

Accepted